# Phytopathogenic Cercosporoid Fungi—From Taxonomy to Modern Biochemistry and Molecular Biology

**DOI:** 10.3390/ijms21228555

**Published:** 2020-11-13

**Authors:** Urszula Świderska-Burek, Margaret E. Daub, Elizabeth Thomas, Magdalena Jaszek, Anna Pawlik, Grzegorz Janusz

**Affiliations:** 1Department of Botany, Mycology and Ecology, Maria Curie-Skłodowska University, Akademicka 19 Street, 20-033 Lublin, Poland; 2Department Plant and Microbial Biology, North Carolina State University, Raleigh, NC 27695-7612, USA; margo@ncsu.edu (M.E.D.); ethomas@ncsu.edu (E.T.); 3Department of Biochemistry and Biotechnology, Maria Curie-Skłodowska University, Akademicka 19 Street, 20-033 Lublin, Poland; magdalena.jaszek@poczta.umcs.lublin.pl (M.J.); anna.pawlik@poczta.umcs.lublin.pl (A.P.); gjanusz@poczta.umcs.lublin.pl (G.J.)

**Keywords:** cercosporoid fungi, *Cercospora s. lat.*, cercosporin, pathogenesis, enzymes, biotechnological application

## Abstract

Phytopathogenic cercosporoid fungi have been investigated comprehensively due to their important role in causing plant diseases. A significant amount of research has been focused on the biology, morphology, systematics, and taxonomy of this group, with less of a focus on molecular or biochemical issues. Early and extensive research on these fungi focused on taxonomy and their classification based on in vivo features. Lately, investigations have mainly addressed a combination of characteristics such as morphological traits, host specificity, and molecular analyses initiated at the end of the 20th century. Some species that are important from an economic point of view have been more intensively investigated by means of genetic and biochemical methods to better understand the pathogenesis processes. Cercosporin, a photoactivated toxin playing an important role in *Cercospora* diseases, has been extensively studied. Understanding cercosporin toxicity in relation to reactive oxygen species (ROS) production facilitated the discovery and regulation of the cercosporin biosynthesis pathway, including the gene cluster encoding pathway enzymes. Furthermore, these fungi may be a source of other biotechnologically important compounds, e.g., industrially relevant enzymes. This paper reviews methods and important results of investigations of this group of fungi addressed at different levels over the years.

## 1. Introduction

Cercosporoid fungi (formerly *Cercospora s. lat.*, *sensu* Chupp, 1954) belong to the Mycosphaerellaceae (Capnodiales, Ascomycota) and are represented by genera and species similar to the genus *Cercospora*. The affiliation of genera within this group of fungi has been problematic for a long time. Sixteen years ago, Crous and Braun [1] distinguished four true cercosporoid genera, i.e., *Cercospora* Fresen., *Passalora* Fr., *Pseudocercospora* Speg., and *Stenella* Syd., as well as several others that are morphologically similar (e.g., *Cladosporiella*, *Phacelium*, *Phaeoisariopsis*, *Stigmina*, *Thedgonia*). Since that time, extensive analysis of phenotypic characters as well as phylogenetic studies have led to a change in this taxonomic concept, and the genus *Zasmidium s. str.* has been assigned to cercosporoids instead of *Stenella* [2,3].

Based on their morphological characteristics and life cycle, the cercosporoids comprise holoblastic asexual morphs, asexual holomorphs, and at least partly mycosphaerella-like sexual morphs [1,3,4]. However, the sexual stage has not been reported for most cercosporoid species, which is associated with the loss of the ability to form sexual morphs (Figure 1) [4,5,6]. Including those that do form sexual morphs, the group should be classified as hemibiotrophs (facultative saprotrophs), i.e., asexual (anamorphic) stages are parasites (biotrophs), while sexual (teleomorphic) forms develop on dead plant remains (as necrotrophs). This indicates that they can complete their life cycles on the dead tissue of leaf spots that they have caused themselves [7].

Cercosporoid fungi represent one of the largest and most important groups of plant pathogenic fungi that cause leaf spots. They comprise economically relevant species causing diseases on a wide range of plants (i.e., dicots, monocots, some gymnosperms, and ferns), including numerous cultivated plants on almost every continent. Although widespread, the cercosporoid species exhibit especially high diversity in tropical and subtropical areas of Africa, Asia, Australia, and Central to South America [3,4]. The most important and often best known and examined diseases caused by cercosporoid fungi include leaf spot of sugar beet (*Cercospora beticola*), celery (*Cercospora apii*), grapevine (*Pseudocercospora vitis*), angular leaf spot of bean (*Pseudocercospora griseola*), black leaf streak of banana (*Pseudocercospora fijiensis*), and leaf spots on many other hosts [4,7].

Symptoms of cercosporoid diseases are usually visible as leaf spots, mostly with a distinct darker margin or they are vein-limited. Within the spots, diagnostic structures (conidiophores and conidia) are produced. These fungi form macronematous conidiophores singly, in fascicles, sporodochia or synnemata, hyaline or pigmented, with terminally or intercalary integrated conidiogenous cells, holoblastic conidiogenesis, percurrent to sympodial proliferation. Conidia are formed singly or in acropetal chains, and they are amero- to scolecosporous and hyaline or pigmented [1,4,7].

Since the genus *Cercospora* was first described, its taxonomy as well as assignment of individual species to this group has been problematic. Although morphological traits are frequently used to identify newly isolated fungi, it is not possible to distinguish *Cercospora* spp. based exclusively on the characteristics of morphological asexual reproductive structures. Enormous taxonomic progress has been made in the last few decades with the application of molecular methods in capnodealean fungi, including Mycosphaerellaceae [5,8,9,10,11]. Symptoms of infection and morphological characters in vivo (characters associated with the fungi on their host plants in nature) are still important in diagnosis of pathogenic fungi, not only for monitoring and identification, but also for understanding their ecology, taxonomy, and economic importance. Molecular techniques are used to overcome taxonomic problems posed by the limitations of morphological characteristics. In turn, the classification of fungi is mainly based on a combination of morphological characteristics, host specificity, and molecular analyses [12,13]. Currently only a limited number of cercosporoid fungi have been cultivated in laboratory conditions, and hundreds of taxa are only known from in vivo analysis and their morphological characters [3]. Therefore, only a limited number of their genes have so far been sequenced and analyzed. The growing number of DNA sequences of cercosporoid fungi provides better insights into the phylogenetic structure of this fungal group at the familial and generic levels. However, there is still a need for further data supplementing current knowledge [3].

In the taxonomic context, the developing molecular analyses have been used in studies of e.g., *Cercospora apii s. lat.* recognized by Crous and Braun [1], and attempts have been made to separate or order the species included therein. The species in this complex that are morphologically indistinguishable from *C. apii s. str.* could not be separated using only the internal transcribed spacers (ITS1 and ITS2) and the 5.8S ribosomal (r)RNA gene [14]. However, using the recent eight-gene analysis (ITS, translation elongation factor 1-alpha (*tef1*) actin (*actA*), calmodulin (*cmdA*), histone H3 (*his3*), β-tubulin (*tub2*), RNA polymerase II gene (*rpb2*), and glyceraldehyde-3-phosphate dehydrogenase (*gapdh*)), it has been possible to separate species grouped in *C. apii s. lat.* that occur on different unrelated plant families [13]. It was first thought that the species within cercosporoids are host specific at the level of the plant genus or family, a concept that led to describing a large number of different species. It is now recognized, for example for *C. apii* Fresen., that some of the species present on diverse hosts are indistinguishable and are the same species [15,16].

## 2. Brief Taxonomical History of Cercosporoids

The genus *Cercospora*, from which the name of the group originates, was established by Fuckel (Fungi Rhen. Exs.: No. 117, 1863; as Fresen. ex Fuckel), and slightly later in the same year Fresenius published a description thereof [17]. He did not give a clear definition of the genus. However, the first genus introduced to the complex of cercosporoid hyphomycetes was *Passalora*, described in 1849 by Fries [18]. Later, Saccardo defined *Cercospora* as having brown conidiophores and brown, olivaceous or sometimes subhyaline, and vermiform conidia. He did not mention *C. apii* with hyaline conidia as a type species, and *Cercospora ferruginea* was considered the typical species of *Cercospora*. At the beginning of the 19th century, Spegazzini [19] was the first to divide the genus and distinguish the genus *Cercosporina*, in which he placed species with hyaline conidia, including *C. apii*. Since that time, the systematic position of *Cercospora* has been changed a few times. At the beginning of the 20th century, the concept of the genus was broad.

The first comprehensive monographic study was published 65 years ago by Chupp [15]. The monograph described 1419 species, published under *Cercospora* and *Cercosporina*. *Cercospora* species were considered to be host specific, and Chupp used this argument to formulate the concept that each plant host genus or family would have its own *Cercospora* species. Additionally, the species concepts and taxonomy were based on morphological characters, such as thickening of hila and pigmentation of single or chained conidia [1,15].

Other significant papers were then issued and discussed by Deighton [20,21,22,23,24,25,26], Pollack [27], Pons and Sutton [28], Braun [29,30,31], and Braun and Melnik [32] based on the diversity of morphological structures. Redefinition of the genus by the authors led to the division of the complex into smaller genera.

In 2003, Crous and Braun undertook a major morphological revision of 5720 names in this group. The number of over 3000 species published in the genus *Cercospora* and 550 in *Passalora* was reduced to 940, of which 281 were placed in the *C. apii* complex as morphologically indistinguishable. They regarded *Cercospora*, *Passalora*, *Pseudocercospora*, and *Stenella* as true genera. The individual genera were distinguished based on a combination of two traits, i.e., the structure of conidial scars and hila and the presence or absence of coloring of conidiophores and conidia [1].

Thus, the generic concept of cercosporoid fungi has been changed many times. Since the description of the first species was published, the application of *Cercospora* has been widened. This broad classification of most species of cercosporoid fungi in the genus *Cercospora* also proposed by Chupp [15] was later divided by other authors into smaller units. This again led to a reduction of the number of recognized genera, as prompted by the first results from phylogenetic analyses using multilocus DNA sequence data (internal transcribed spacers, actin, calmodulin, histone H3, and translation elongation factor 1-alpha genes) [1,4]. A detailed history of the group has been published many times, e.g., by Deighton [24], Braun [31], Crous and Braun [1], and To-Anun et al. [33].

Lately, a monographic series of articles about cercosporoid fungi successively revising the representatives of various groups of plants (e.g., Pteridophyta, Gymnospermae, monocots, dicots) has been published [4,34,35,36]. The articles present both traditional descriptions in vivo and characteristics of colonies in vitro of revised species. It should be mentioned that taxonomic research on this group is still being carried out and required.

An important part of research comprises investigations of Mycosphaerellaceae phylogeny and most assumptions (morphological features mentioned above and preliminary molecular studies cited) proposed by Crous and Braun [1] were later confirmed by molecular analyses [4,5,8,37,38,39]. The genus *Cercospora* is considered to be monophyletic [5,9], while the others, i.e., *Pseudocercospora* [8], *Passalora* [4,40], and *Stenella* [5,41] are regarded as polyphyletic within the Mycospharellaceae.

## 3. Molecular Investigations of Cercosporoids

Molecular studies, which have become an integrated element in taxonomic research, are an important complementation to diagnosis and morphological descriptions of fungi. Significant changes within cercosporoid genera have been demonstrated by means of molecular analyses, including phylogenetic studies, which have been developing intensively since the first years of the 21th century. Various genetic techniques have been used for *Cercospora* identification, i.e., partial sequences of genes encoding HIS, TEF-1α, TUB, ACT, and CAL as well as the 18S rRNA gene and the ITS regions that include the 5.8S rRNA gene (ITS-5.8S-ITS2) [6,14,42,43,44]. Recently, molecular analyses using the six genes mentioned previously have been extended to include additional two, i.e., RNA polymerase II and glyceraldehyde-3-phosphate dehydrogenase genes, which facilitated separation of some fungal species from *C. apii s. lat.* [13].

To elucidate the probable genetic diversity in the cercosporoids, most of the papers published so far were focused on sequencing of single genes as mentioned above (ITS, LSU, tef1, actA and others) [6,13]. These papers are of great scientific value as they not only facilitate arrangement of the taxonomy of this group, but also give an idea about fungal migration, ecology, and even reproduction. However, genomic analysis based on next-generation sequencing (NGS) across different fungal taxa gives more information on their probable evolution, development, and infection strategies. Zeng et al. [45] presented a comparison of the *Cercospora sojina* genome with genomes of plant pathogen members of the genus *Mycosphaerella*, i.e., *M. graminicola* (currently *Zymoseptoria graminicola*, *M. pini* (currently *Dothistroma pini*), *M. populorum* (currently *Sphaerulina musiva*), and *M. fijiensis Pseudocercospora fijiensis* (currently *Pseudocercospora fijiensis)*. The authors suggested that repetitive elements may be responsible for considerable evolutionary changes, and they elucidated genes involved in pathogen specialization, such as pathogenicity to woody vs. herbaceous hosts. It was shown that *C. sojina* lacks genetic elements that function as transcription repressors and for subtilases that act as “chemical weapons”. In contrast, the fungus has a significantly high number of genes coding for proteins involved in cell division and oxidation reduction [45].

A substantial part of research on this genus analyzes its ability to infect agriculturally important plants. Therefore, different techniques allowing detection of fungal presence in the plant material have been developed, as successful evaluation of the infection scale is crucial in agronomy and in view of a possible further use of infected plants for feeding animals or humans. For example, real-time PCR was applied for the quantification of *C. beticola* pathogen biomass in the infected beet canopy [46]. Since the full virulence of these fungi essentially depends on cercosporin, much effort has been dedicated to elucidating its synthesis [47]. Moreover, it should be mentioned that, being toxic to all cells, cercosporin may potentially be hazardous to its own producer. Studies have suggested that as many as 185 genes may be engaged in resistance of *Cercospora nicotianae* to this toxin, and may include genes encoding reductases, antioxidants and reactive oxygen species (ROS) quenchers, and membrane transporters [48].

The recent explosion of papers describing fungal genomes and transcriptomes have allowed researchers to perform more detailed analyses of fungal biology and ecology. By sequencing the *C. sojina* genome, Luo et al. [49] identified approximately 750 secreted proteins including 141 putative pathogen effectors. Interestingly, only one CBM1 (carbohydrate binding module) protein was found in the *C. sojina* genome. The authors suggested that considering the relatively slow infection by the fungus, the deficiency of CBMs in the genome may weaken *C. sojina* infection in terms of digesting plant cell walls [49]. Moreover, this fungus was shown to lack cercosporin, and it is suspected to produce another mycotoxin that facilities the infection process. It should be emphasized that the use of transcriptomics to analyze the infection process may pose difficulties because the progress of infection varies in time. Possibly, each *Cercospora* species should be considered separately. Thus, Luo et al. [49] suggested that starvation treatments may mimic fungal physiology during infection, and they performed transcriptome analysis from nutrient-limited 24-h and 48-h in vitro cultures. They showed that over one thousand genes were differentially expressed and therefore are potentially involved in pathogen–host interactions. These included 260 genes encoding secreted proteins, including 81 putative effectors, almost 200 genes coding for CAZymes (Carbohydrate-Active Enzymes), and 21 genes involved in synthesis of secondary metabolites. Additionally, detailed analysis of maize infection by *C. zeae-maydis* from Bluhm et al. [50] proved that engagement of several genes in development of this fungus depended on lighting conditions. This observation may further complicate our understanding on how *Cercospora* spp. infects plants.

## 4. Cercosporin

As noted above, much of the success in the pathogenesis of *Cercospora* species has been attributed to their production of a highly toxic, photoactivated metabolite called cercosporin. Cercosporin was first isolated in 1957 and investigated as an interesting pigment due to its red color that is easily visible in mycelial cultures [51]. The perylenequinone structure and chemical properties of the compound were determined in 1972 [52]. Its role as a toxin and as a photoactivated compound was not reported until 1975 when Yamazaki et al. demonstrated cercosporin’s toxicity to mice and bacteria and the dependence of toxicity on light and oxygen [53]. Many subsequent studies have documented its toxicity to plants [54,55,56].

Cercosporin has been of significant interest as it was the first plant pathogen toxin to be shown to be a photosensitizer, a compound that is activated by light to produce reactive oxygen species toxic to living cells (Figure 2) [57]. Photosensitizers are documented to damage many cellular components including lipids, proteins, and nucleic acids, with the site of damage dependent on where the photosensitizer molecule localizes in cells, such as membranes, the cytoplasm, or nucleus [58].

### 4.1. Role in Disease

Cercosporin toxicity to plant tissue results from the breakdown of cellular membranes caused by peroxidation of membrane lipids [61,62]. Evidence for an important role for cercosporin in disease comes from many studies. Research in coffee and sugar beet have shown a reduction of symptoms and increased time to symptom expression when leaves are shaded, consistent with the requirement for light activation for cercosporin activity [63,64]. In addition, cercosporin can be isolated from lesions on infected plants, and ultrastructural studies show membrane damage consistent with cercosporin’s mode of action [65,66].

The most compelling evidence for the role of cercosporin comes from studies documenting reduction in infection and disease severity by mutants deficient in cercosporin production. These studies were built on the groundbreaking work of Chung and co-workers, who identified the gene cluster encoding the cercosporin biosynthetic pathway [56,67,68,69]. Characterization of cercosporin-deficient mutants created by restriction enzyme-mediated insertion (REMI) identified a gene encoding a polyketide synthase (PKS) required for cercosporin production [67]. Subsequent flanking sequence analysis led to the identification of a cluster of eight polyketide synthase genes (*CTB1*–*CTB8*) in the pathway; these encode one polyketide synthase, two methyltransferases, three oxidoreductases, a major facilitator superfamily (MFS) transporter, and a zinc finger transcription factor [68]. Identification of this cluster led to characterization of the biosynthetic pathway and metabolic intermediates [70], as well as to a recent genomic analysis that has identified five additional loci flanking the CTB cluster that are involved in synthesis in *C. beticola* [71]. One of these loci encodes the cercosporin facilitator protein (CFP), an MFS transporter previously shown to play an important role in cercosporin synthesis and autoresistance in *Cercospora kikuchii* [72]. The ABC transporter gene, *ATR1*, has also been shown to play a major role in cercosporin production [73].

The identification of cercosporin biosynthesis genes has been used to define the requirement for cercosporin in disease development. *C. nicotianae* and *C. beticola* mutants deficient in the CTB1 PKS both produced fewer and smaller lesions when inoculated onto tobacco and sugar beet, respectively [67,69]. Mutants for the CTB4 MFS transporter in *C. nicotianae,* the CTB2 O-methyltransferase in *C. beticola*, and the CFP MFS transporter in *C. kikuchii* were also shown to be less pathogenic when inoculated onto their hosts [72,74,75]. In addition, mutants for the CZK3 MAP kinase kinase regulator required for cercosporin production in *Cercospora zeae-maydis*, produced only small chlorotic spots when inoculated on corn [76]. These studies document a critical role for cercosporin in disease development.

### 4.2. Cercosporin Production by Cercospora Species

Cercosporin production has been documented in a wide range of *Cercospora* species pathogenic on diverse hosts. Three independent studies in the 1970s documented production in 7 out of 12 species, in 24 of 61 species, and in 12 out of 20 species assayed, respectively [65,77,78]. It has been noted that some of the species investigated in these early studies are no longer classified as *Cercospora* species due to more recent phylogenetic and taxonomic analyses [79]. Thus, the actual proportion of cercosporin producers in the genus is likely higher than the studies indicate. Species commonly known to produce cercosporin include *C. beticola, C. kikuchii, C. nicotianae, C. zeae-maydis, C. coffeicola, C. canescens, C. zebrina, C. malvicola, C. personata, C. hayii, C. apii, C. sorghi,* and *C. asparagi* [6,65,78,79,80].

The lack of cercosporin production in screening studies should not be taken as definitive evidence that isolates of a particular species do not synthesize cercosporin, as conditions that induce production in culture vary with species and isolate. A study of four different isolates of *C. kikuchii* along with isolates of *C. beticola, C. nicotianae, C. asparagi,* and *C. zeae-maydis* showed that production was affected by medium composition, with differential effects of specific substrate components on different species and isolates within a species [81]. Production by some, but not all, isolates was impacted by the carbon:nitrogen ratio in the medium. Isolates were similar in producing more cercosporin when incubated in the light and when grown at 20 °C instead of 30 °C, but differed in their specific responses. A subsequent extensive study by You et al. [82] identified significant variation in cercosporin production on different brands of potato dextrose agar, and in the presence of various micronutrients, buffers, carbon and nitrogen sources, and metal ions, with different species responding differently to medium conditions. In the case of *C. nicotianae*, conditions that stimulate conidiation suppress cercosporin production [56].

Several studies have documented the lack of production by some species of *Cercospora.* Two distinct species of *Cercospora, C. zeae-maydis* and *C. zeina* cause the Gray Leaf Spot disease of corn [79]. The two species have both distinct and overlapping distributions and cause similar symptoms. Only *C. zeae-maydis*, however, has been shown to synthesize cercosporin. Similarly, although *C. kikuchii* is a prolific cercosporin producer, *C. sojina*, also a soybean pathogen, has not been shown to produce cercosporin [6,79]. Interestingly, a recent genomics study identified a gene cluster in *C. sojina* with high homology to the CTB cercosporin biosynthesis cluster, although the authors were not able to identify cercosporin from *C. sojina* either in culture or in planta [49]. However, further detailed studies comprising genomics, proteomics, and biochemistry would be useful in explanation of such phenomena.

Although some phylogenetic studies have suggested that cercosporin production is limited to species within the genus *Cercospora* [6,79], some investigations have demonstrated production in other genera, primarily those found in the class Dothideomycetes, which are the most active producers of photoactivated perylenequinones [56]. Cercosporin producers include isolates of *Pseudocercosporella capsellae*, a closely related genus in the *Mycosphaerella* group [83], as well as *Stagonospora* [84]. A recent report documented cercosporin production by *Colletotricum fioriniae* [71], a genus not previously reported to produce any perylenequinone toxin, although members of other genera in the class Sordariomycetes, such as *Hypocrella* and *Hypomyces*, have been shown to produce photoactivated perylenequinones [56]. These studies suggest that cercosporin production is more widespread than previously thought, and thus may be of more ancient origin.

### 4.3. Cercosporin Resistance

The importance of cercosporin in many *Cercospora* diseases has led to studies to identify mechanisms to engineer resistance by targeting the toxin itself. As mentioned above, cercosporin shows almost universal toxicity, not only to plants, but to mice, bacteria, and fungi. This almost universal toxicity is due to the toxin’s production of reactive oxygen species such as singlet oxygen and superoxide (see section below on redox processes). Thus, a major approach has been to identify cercosporin autoresistance genes in the fungus as a possible source of resistance genes for engineering resistance in host plants [85].

Early physiological studies focused on known quenchers of ^1^O_2_ and other antioxidant defenses in cells. Carotenoids are among the most effective of ^1^O_2_ quenchers in biological systems, but carotenoid-deficient mutants of *C. nicotianae* retained normal resistance to cercosporin [86]. Studies of other antioxidant cellular defenses such as thiols, reducing substrates, and antioxidant enzymes also showed no role [85]. The major mechanism of resistance identified resulted from microscope observations that cercosporin-producing mycelium emitted green rather than normal red fluorescence, suggesting that the compound was modified when in contact with fungal hyphae. Studies on the fluorescent properties and lack of toxicity of reduced derivatives of cercosporin, along with studies on the cell-surface reducing activity of *Cercospora* relative to cercosporin-sensitive fungi have all supported the hypothesis that *Cercospora* fungi are resistant due to the ability to retain cercosporin in contact with the cell in a reduced and non-toxic state [85,87,88,89].

Potential cercosporin-resistance genes were initially identified through complementation of cercosporin-sensitive mutants. Several mutants were deficient in pyridoxine (vitamin B_6_) synthesis, and subsequent work showed that pyridoxine and its vitamers were effective quenchers of ^1^O_2_ [85,90,91]. Another mutant was deficient in a zinc cluster transcription factor (CRG1), which was shown to be involved in both cercosporin resistance and production [92]. Subtractive hybridization between the wild type and the cercosporin-sensitive *crg1* mutant led to the identification of differentially regulated genes, some of which were hypothesized to be responsible for cercosporin autoresistance [93]. Selected genes from the subtractive library have been characterized for expression under cercosporin-toxicity conditions and for the ability to impart resistance to the cercosporin-sensitive fungus *Neurospora crassa.* Of these, two genes, one encoding an ABC transporter (ATR2) and one encoding a hypothetical protein (71cR) were shown to impart cercosporin resistance when expressed in *N. crassa* and are possible targets as resistance genes for crop improvement [48,94]. Studies were also conducted with an additional six of the library genes that were homologous to ^1^O_2_-resistance genes characterized in the photosynthetic bacterium *Rhodobacter sphaeroides*, a model for understanding cellular ^1^O_2_ resistance [95]. None of the *Rhodobacter* homologs were found to impart cercosporin resistance in *N. crassa*, however.

## 5. Importance of Redox Processes in Cercosporoid Physiology

In the context of growth and pathogenesis mechanisms observed in fungi belonging to *Cercospora*, a life strategy associated with cell redox mechanisms seems to be very important. An efficient redox system is essential, both during the attack of the pathogen on the plant cells and its natural life cycle associated with the synthesis of cercosporin [89].

The efficient production of ROS is a part of specific plant defense response mechanisms to various types of stress factors [96]. The attack of a pathogen is one of the mechanisms triggering a broad spectrum of responses to reduce or completely inhibit the growth of the pathogenic agent. In response to microbial infection, plants carry out, e.g., a rapid and transient synthesis of a large amount of free radicals, such as superoxide anion radical (^•^O_2_^−^), hydroxyl radical (^•^OH), and hydrogen peroxide (H_2_O_2_), as a rapid signaling pathway. This phenomenon is called the oxidative burst [97]. The amounts of ROS produced by plant tissues are highly variable and are in the range of concentrations from nM to mM [98,99,100]. Organisms, including fungi, have developed complex antioxidative protection mechanisms against oxidative stress based on both enzymatic and non-enzymatic molecules [101,102]. The main enzymatic antioxidants that are able to protect cells in oxidative stress conditions are superoxide dismutase (E.C. 1.15.1.1), catalase (EC 1.11.1.6), or peroxidase (EC 1.11.1.7). Modern molecular techniques, including genomic sequencing, can currently help to accurately identify each enzyme based on, e.g., genes encoding thereof. These enzymes are also produced by fungi belonging to the genus *Cercospora,* where they are responsible for the scavenging of superoxide anion-radicals or the decomposition of hydrogen peroxide. The presence of superoxide dismutase activity in the extracts of *C. nicotianae* mycelia has already been shown (43% of superoxide reduction in the pH 7.8). Similarly, the activity of catalase and peroxidase has also been described for this fungus [103].

In the case of cercosporoid fungi, the photosensitizer connected with the pathogenesis in plants, i.e., cercosporin, absorbs light and is then converted to an activated “triplet” state. In this activated state, it may react directly with oxygen to produce highly reactive singlet oxygen (^1^O_2_). It may also react through a reducing substrate to produce a reduced sensitizer molecule that can damage cells by reacting directly with cellular molecules or by generating free-radical forms of oxygen such as superoxide (^•^O_2_^−^) or the hydroxyl radical (^•^OH) [57,58,90,104]. Cercosporin has been shown to produce both ^1^O_2_ and ^•^O_2_^−^, but its toxicity has been primarily attributed to the production of ^1^O_2_ due to its high quantum yield and the ability of ^1^O_2_ quenchers to protect against cercosporin toxicity [105,106,107].

As strong oxidizers, ROS, such as ^1^O_2_, ^•^O_2_^−^ or ^•^OH, may initiate a series of oxidative reactions in various biological systems. Organisms producing toxins that generate these ROS have evolved mechanisms of resistance, which facilitate undisturbed biomass growth even in the presence of mM concentrations of the toxins [87]. As noted above, enzymatic antioxidant defenses were not shown to play a major role in cercosporin resistance. Superoxide dismutase, catalase, and peroxidase activities in *C. nicotianae* species were not adequate to provide protection [103]. In terms of general antioxidant defenses, however, the addition of reducing agents such as glutathione, ascorbate, or cysteine to cultures of cercosporin-sensitive fungi significantly reduced its toxicity [89]. The antioxidant vitamin B_6_ (pyridoxine) was shown to have the greatest role in cercosporin resistance. Pyridoxine and its derivatives produced by *Cercospora* species can quench both ^1^O_2_ and the activated (triplet) form of the toxin to provide resistance [91]. The reductive detoxification of cercosporin, mentioned above, may also be an indication of redox defenses [87,89,108]. The cercosporin observed in hyphae of resistant species like *C. kikuchii* or *C. nicotianae* was in a reduced state and was reoxidized after diffusion from the fungal cell surface. In contrast, in cercosporin–sensitive fungi, the toxin was present in its oxidative state. Assays of the reduction of 20 tetrazolium dyes differing in redox potential showed that fungi resistant to cercosporin could reduce more compounds than sensitive species [89]. The redox system of fungi belonging to *Cercospora* has become the target of modern plant protection. It has been described that the Fenton solutions, e.g., Fenton’s reagent (a mixture of hydrogen peroxide and Fe^2+^ ions), Fenton-like reagents (a mixture of hydrogen peroxide and Fe^3+^ ions), or a new type of Fenton’s reagent (hydrogen peroxide, Fe^3+^ ions, and oxalic acid) are applied onto sugar beet to control the leaf spot induced by *C. beticola* [100].

## 6. Enzymes Involved in the Plant-Pathogen Interaction

Biochemical properties of this group of microorganisms have been examined so far in terms of the metabolic response to plant infection and induction of pathogenicity. Plants respond to pathogen infection by induction of several genes coding for proteins with a potential role in defense, which are synthesized as a result of a plant’s hypersensitive immune response. In general, plant pathogens produce a wide range of enzymes that are capable of plant cell wall degradation during infection. A number of cellulases, chitinases and pectinases, as well as lignin-modifying enzymes (LME) have been isolated and their biological role has been examined [109,110,111,112,113].

Most *Cercospora* species produce low-molecular-weight phytotoxins and hydrolytic enzymes that weaken and damage cells during their growth and plant invasion. In addition to the production of singlet oxygen by cercosporin [54], which acts as a virulence factor, the mechanisms relevant for plant-fungal interactions comprise pathogenic enzymes secreted by the *Cercospora* species. They are considered as additional potential virulence factors and include esterases, cellulases, and pectinases [112,114], which are known for their potential application in the wood industry and in biofuel production [115,116]. It has been shown that phytopathogenic isolates of *Pseudocercospora ocellata* (=*Cercospora theae*) produce various carbohydrases, pectinase, protease, and amylase during attack on tea plants, suggesting that these enzymes may be imparted to leaves during infection and can be involved in induction of pathogenesis and penetration of plant material [112]. The ability to produce cellulolytic (CMCase) and pectolytic (polygalacturonase) enzymes has also been demonstrated in the case of *Cercospora arachidicola* [117]. In turn, light-dependent synthesis of esterase has been showed in liquid cultures of *C. beticola* [118].

## 7. Biotechnological Applications of Cercosporoid Fungi

The majority of applied research on cercosporoids is related to the toxin produced by these fungi. Cercosporin has been previously linked to its phototoxicity connected with the ability to generate ROS in the presence of light [54]. It was very important to understand the mode of action of the toxin and the mechanisms of resistance in the pathogenesis of host plants, due to the diseases induced and thus losses in crops. Only two of the characterized putative cercosporin-resistance genes have so far been tested in transgenic plants. Transformation of tobacco with the *C. nicotianae* pyridoxine biosynthesis genes *PDX1* and *PDX2* failed to increase cercosporin resistance [119]. Analysis of the transgenic plants showed that expression of the transgenes did not increase pyridoxine vitamer levels due to the down-regulation of the endogenous tobacco PDX genes, suggesting that elevated B_6_ vitamin levels were detrimental to the plants. Transformation of tobacco with other potential cercosporin-resistance genes is currently in progress.

Cercosporin has also been reported as a compound with antibiotic properties. It was active against Gram-positive bacteria and exhibited low antifungal activity [120]. Antileishmanial, moderate antiplasmodial, and cytotoxic activities of cercosporin isolated from culture medium of *Septoria pistaciarum* have recently been found [121]. There is a need to continue similar studies on this metabolite isolated from cercosporoids.

In recent years, the biomedical use of fungi has been more intensively studied. Perylenequinones have been studied for some time for photodynamic therapy of cancer (PDT) in humans, but only recently was cercosporin tested. The photocytotoxicity of cercosporin against two glioblastoma multiforme and one breast adenocarcinoma human cell lines were investigated. It was established that the toxin was suitable for superficial PDT treatments, especially in poorly accessible cancers [122].

Cercosporoid fungi may be a source of other biotechnologically important compounds and substances, including industrially relevant enzymes; however, very few *Cercospora* species have been reported as enzyme producers (Table 1). Nonetheless, partial purification and biochemical characterization of lipase from *C. kikuchii* has been performed including enzyme stabilization studies with the spray drying technique [123,124]. The enzyme was also subjected to different immobilization strategies using chitosan microparticles, and appeared to be suitable for industrial applications [125]. Moreover, laccase from *Cercospora* SPF-6 has been identified recently and used for dye decolorization after immobilization procedure [126].

## 8. Conclusions

Cercosporoid fungi are an important group of parasites known mostly for their pathogenic role on plants. Throughout over 150 years of investigations on cercosporoids, the systematic position of these fungi has been changed, and numerous cercosporoid species have been redefined, mainly based on their morphological characters in vivo. Traditional methods of identification, extended and supplemented in recent decades with molecular techniques, are still needed and are absolutely essential for the description of new species and for phylogenetic studies. Conclusions based solely on morphological analyses are unreliable and do not allow proper generic allocations. It is expected that further research in this field may bring further changes.

Fungal systematics are an essential part of studies on cercoporoids, especially due to their economic and ecological importance. Additionally, given the losses caused to crops, cercosporoids have been subjected to biochemical and molecular research to elucidate the strategies of their pathogenicity. In all cases, however, studies have focused on a single species. In general, the knowledge on cercosporoid enzymology was acquired with simple analytical methods; hence, it should be supplemented using the newest biochemical techniques and tools to reveal the entire biotechnological potential of these microorganisms. The greatest attention has been devoted to research on the mechanism of cercosporin toxicity and cellular defense mechanisms and on developing strategies for blocking production of this toxin. More studies towards detailed knowledge of not only enzymes secreted by cercosporoids and necessary for colonization of the host but also those secreted in oxidative stress conditions should be conducted.

Further metabolic studies are also needed. Cercosporoids can be subjected to metabolic profiling using the Biolog system as well as determination of chemical sensitivity in the presence of the analyzed compound. This will allow identification of factors that may limit the spread of these organisms in their natural environment. Furthermore, antibiotic properties and cytotoxic activities of cercosporin, also reported for representatives from other genera, can be checked in cercosporoids. Recent reports on cercosporin in medical research, for example its photocytotoxicity against cancers, indicate a new direction for applied research. Research to identify new fungal bioactive fractions with potential biomedical application would be of significant importance.

## Figures and Tables

**Figure 1 ijms-21-08555-f001:**
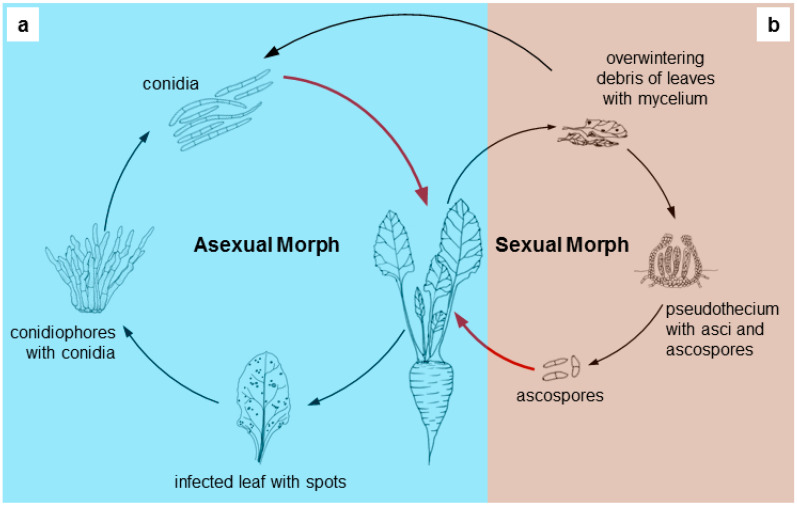
Scheme of the life cycle of cercosporoid fungi: (**a**) blue part—asexual; (**b**) brown part—sexual (if occurs). Red arrows—infection process.

**Figure 2 ijms-21-08555-f002:**
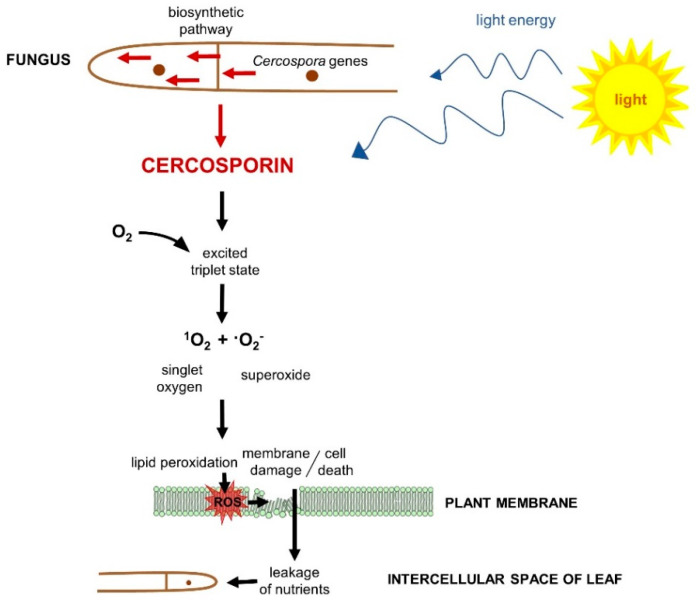
The scheme of cercosporin action (adopted from Daub and Ehrenshaft [54], Daub and Chung [59], and Fuller et al. [60]).

**Table 1 ijms-21-08555-t001:** Enzymes from cercosporoids and their potential biotechnological application.

Enzyme	Source (Species Name) ^a^	Citations	Potential Biotechnological Application
lipase	*Cercospora kikuchii*	[123,124,125]	food and chemical industry, biodiesel transesterification
esterase	*C. beticola*	[118]	wood industry and biofuel production
cellulases	*C. beticola*	[115,127]	wood industry and biofuel production
pectinases	*C. beticola*	[115,127]	food and textile industry
laccase	*Cercospora* SPF-6	[126]	dye decolorization
endo-glucanase (CMCase)	*C. theae*	[112]	wood industry and biofuel production
invertase	*C. theae*	[112]	food industry, cosmetics and pharmaceutical industry
protease	*C. theae*	[112]	chemical and textile industry, food industry
pectinase	*C. theae*	[112]	food and textile industry
amylase	*C. theae*	[112]	food and textile industry, pharmaceutical and fine-chemical industry
CMCase	*C. arachidicola*	[117]	food industry and biofuel production
polygalacturonase (pectinase)	*C. arachidicola*	[117]	food and textile industry

^a^ Fungal names in the table were originally published in the papers. The current name of *C. thea* is *Pseudocercospora ocellata* (acc. to IndexFungorum), while *C. arachidicola* should be classified as *Passalora arachidicola* [128].

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
