# Peer review of "Phytopathogenic Cercosporoid Fungi—From Taxonomy to Modern Biochemistry and Molecular Biology"

_ijms, 2020, doi:10.3390/ijms21228555_

Round 1
Reviewer 1 Report
The paper entitled “Phytopathogenic Cercosporoid Fungi - From Classical Taxonomy to Modern Biochemistry and Molecular Biology” is a well written review about a significant group of phytopatogenic fungi. Generally, the aim of the paper was not well presented, but I suggest the publication in IJMS after minor revision in paper organizzation.
Abtrasct: the abstract need to be refocused. In the first part, it seems that the review will consider the taxonomic evolution of cercosporoid fungi. Later, the biochemical implication of cercosporin is introduced in relation to disease development, or the searching of useful chemicals. Thus, it is not really clearly reported the aim of the review (see also comment at 171, 201). I think that most of the contents are suitable to retrace the history of this group of fungi in its entirety, but the title, abstract and some parts seem to suggest, instead, a treatment of the evolution of the taxonomic approach.
L. 75. Please use Italic for “in vivo”. Please check in the whole manuscript.
L.171-200. I think that this part cannot be included in this form into “molecular investigations of cercosporoids”. I know that most diagnostic methods are molecular one, but I think that this chapter is a sort of counter part of the previous one, in which taxonomy was driven by morphology. So, this chapter should be limited to molecular approach to taxonomy. Maybe this part should be a stand-alone one, in which other molecular approaches were investigated and reported here, but Author should find a better link to the taxonomic approach of the first part of the review.
L. 201. As above, it is not clear the sense of reporting cercosporin research in this review. Is cercosporin related to taxonomy? Or Authors just reports the whole mass of studies about this group of pathogens?
Author Response
Response to Reviewer 1 Comments
Point 1. Abtrasct: the abstract need to be refocused. In the first part, it seems that the review will consider the taxonomic evolution of cercosporoid fungi. Later, the biochemical implication of cercosporin is introduced in relation to disease development, or the searching of useful chemicals. Thus, it is not really clearly reported the aim of the review (see also comment at 171, 201). I think that most of the contents are suitable to retrace the history of this group of fungi in its entirety, but the title, abstract and some parts seem to suggest, instead, a treatment of the evolution of the taxonomic approach.
Response 1: AbstractThe word ‘phylogenetic’ suggesting that it contains taxonomic evolution has been deleted and replaced by ‘molecular’ – which is included in the manuscript. The last sentence has been changed to emphasize the content of the work, which includes a review of the most important research methods on cercosporoid fungi and their most important results in various fields, with an emphasis on the aspect of pathogenicity in the context of future perspectives, i.e. biotechnological application.
The title, briefly and simply, is to show that it concerns a review of the above-mentioned areas of research over the years.
Point 2. L. 75. Please use Italic for “in vivo”. Please check in the whole manuscript.
Response 2: I have used Italic for ”in vivo” and „in vitro” in the whole manuscript.
Point 3. L.171-200. I think that this part cannot be included in this form into “molecular investigations of cercosporoids”. I know that most diagnostic methods are molecular one, but I think that this chapter is a sort of counter part of the previous one, in which taxonomy was driven by morphology. So, this chapter should be limited to molecular approach to taxonomy. Maybe this part should be a stand-alone one, in which other molecular approaches were investigated and reported here, but Author should find a better link to the taxonomic approach of the first part of the review.
Response 3: The part “molecular investigations of cercosporoids” covers various aspects of research at the molecular level, also partially with regard to taxonomy. It shows tests for pathogenicity, the ability to infect plants, the course of the infection process, and resistance to the toxin produced. This chapter should remain separate.
Point 4. L. 201. As above, it is not clear the sense of reporting cercosporin research in this review. Is cercosporin related to taxonomy? Or Authors just reports the whole mass of studies about this group of pathogens?
Response 4: The paper presents a review of the most important research on cercosporoid fungi in various areas - taxonomy, biochemistry, and genetics in plant-parasite interactions. The cercosporin research review emphasized a role in plant diseases caused by cercosporoids. The latest important and detailed review about cercosporin was published almost 20 years ago and here it has been completed.
Reviewer 2 Report
Dear authors,
As a whole, it is very interesting review article, but the title is so large.
You described from taxonomy to modern biochemistry and molecular biology.
But I think that you should mention taxonomy shorter, and describe in detail about cercosporin and its biosynthesis, cercosporoid physiology, enzymes involved in the plant-pathogen interaction.
If you would describe taxonomy, you should refer Braun et al. 2013 ; Groenewald et al. 2013; Videira et al. 2017. Those are in “Studies in Mycology”, “IMA Fungus”.
L 38 Crous and Braun [1] → I cannot find it in “References”.
L 144 you refer [40] [41] → I think that are not appropriate. Please reconsider.
L 430 In table 1, I think that citations do not fit “potential biotechnological application”.
From citations, I cannot read the “potential biotechnological application” like wood industry, biodiesel and biofuel.
With best regards,
Author Response
Response to Reviewer 2 Comments
Point 1. As a whole, it is very interesting review article, but the title is so large.
Response 1: The title has been shortened. The word 'classical' has been removed because the paper deals with the molecular aspect of the taxonomy.
Point 2. You described from taxonomy to modern biochemistry and molecular biology. But I think that you should mention taxonomy shorter, and describe in detail about cercosporin and its biosynthesis, cercosporoid physiology, enzymes involved in the plant-pathogen interaction.
Response 2: The review covers only the most important information in the context of the evolution of research and studies of cercosporoid fungi with the plant-pathogen aspect. The detail history of the research has been described several times - the references are listed in the article and the most important information is provided here.
Point 3. If you would describe taxonomy, you should refer Braun et al. 2013 ; Groenewald et al. 2013; Videira et al. 2017. Those are in “Studies in Mycology”, “IMA Fungus”.
Response 3: This very valuable and extensive paper on cercosporoid fungi was previously included in the work, but due to technical problems with the EndNote program, part of the bibliography was not included in the list. Hence, there is a discrepancy between the numbers and the list, but now everything has been carefully checked and corrected.
Point 4. L 38 Crous and Braun [1] → I cannot find it in “References”.
L 144 you refer [40] [41] → I think that are not appropriate. Please reconsider.
Response 4: As above – problems with EndNote, which have been corrected.
Point 5. L 430 In table 1, I think that citations do not fit “potential biotechnological application”.
From citations, I cannot read the “potential biotechnological application” like wood industry, biodiesel and biofuel.
Response 5: The quotes refer to enzymes and not to potential biotechnological use. Therefore, the column with potential use follows the citation.
Round 2
Reviewer 2 Report
Dear authors,
You accurately corrected.
But a few points of mistake are remained.
L 136 To-Ann[33] → To-Ann et al.[33]
L 152 20th century → 21th century
Best regards,